# The Larger the Better? Improved LLM Code-Generation via Budget Reallocation

**Michael Hassid**[1,2]*, **Tal Remez**[1]*, **Jonas Gehring**[1], **Roy Schwartz**[2], **Yossi Adi**[1,2]
[1]FAIR Team, Meta
[2]The Hebrew University of Jerusalem
{michael.hassid}@mail.huji.ac.il

## Abstract

It is a common belief that *large* language models (LLMs) are better than smaller-sized ones. However, larger models also require significantly more time and compute during inference. This begs the question: *what happens when both models operate under the same budget?* (e.g., compute, run-time). To address this question, we analyze code generation LLMs of various sizes and make comparisons such as running a 70B model once vs. generating five outputs from a 13B model. We consider a standard unit-test setup, which can be used to select the correct output from the smaller model. Our findings reveal that the repeated use of smaller models can yield consistent improvements, with gains of up to 15% across five tasks. On the other hand, in scenarios where unit-tests are unavailable, a ranking-based selection of candidates from the smaller model falls short of the performance of a single output from larger ones. Our results highlight the potential of using smaller models instead of larger ones, and the importance of studying approaches for ranking LLM outputs.[1]

## 1 Introduction

A common wisdom in deep learning, and language modeling in particular, is that investing more compute leads to improved performance (Kaplan et al., 2020). The standard way of implementing this principle is training larger models. A simpler, yet often overlooked way to increase compute budget is to run a smaller model multiple times, and select the best output using some metric (Chen et al., 2021). In this work we systematically compare between these two approaches: we ask whether, given a fixed compute budget, it is best to run a large model once, or a smaller model multiple times (Figure 1). Our results show that, perhaps surprisingly, given the same compute budget, running 7B or 13B models can not only match the performance of a 70B model, but also substantially surpass it.

Addressing our research question requires a method for selecting the best LLM output from a given set of candidates. In this work we focus on execution based code-generation tasks, which assume the availability of unit-tests (Chen et al., 2021; Austin et al., 2021; Hendrycks et al., 2021). We consider the widely-used pass@*k* metric (Kulal et al., 2019), which evaluates a model's performance on code generation problems by generating *k* outputs and assigning a point if any of them passes all tests. To adapt this metric for our purposes, we take models of different sizes, and for each generate as many outputs as possible given a fixed compute budget, e.g., floating point operations (FLOPs) or wall-time.

We apply this setup to evaluate the Code Llama (Roziere et al., 2023) model family (7B, 13B, 34B, and 70B) across five tasks: HumanEval (Chen et al., 2021), MBPP (Austin et al., 2021), and the three splits of APPS (Hendrycks et al., 2021). For the HumanEval and MBPP benchmarks, we additionally use the recent Llama-3 (AI@Meta, 2024) model family (8B and 70B). Surprisingly, we find that for the two popular tasks, HumanEval and MBPP, the

---

* Equal contribution

[1] Data is avalible at https://github.com/slp-rl/budget-realloc

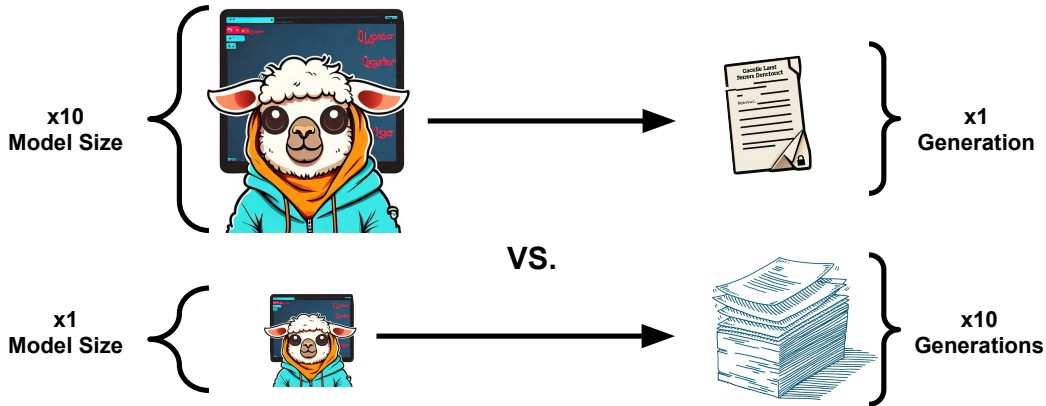

Figure 1: Different ways to improve LLM performance by increasing compute budget. Top: the standard approach of increasing model size, while generating a single output. Bottom: our approach—using a small model to generate multiple outputs, and select the best one.

smaller models (7B, 8B and 13B) outperform the larger ones (34B and 70B) by a margin of up to 15%. Importantly, this is observed using both budget types (FLOPs and wall-time) and across all computation budgets. When considering the challenging APPS benchmark, we find that the 13B model performs best across almost all budgets, with a consistent margin of 5% when considering the hardest split—competition.

We then proceed to examine the scenario where unit-tests are *unavailable*, such as in an IDE code-completion setup. In such cases, an efficient policy is required to select a single solution from all generated ones. We consider a simple LLM-based policy, which ranks solutions based on the negative log likelihood of the LLM. We also augment this policy with a variant of a recent ranking approach—LEVER (Ni et al., 2023). We experiment with the 7B model, and rank its outputs using each of the models. Our results show that, as expected, ranking-based selection improves with the increase in compute budget, and with the size of the ranking LLM. Nonetheless, this procedure still falls short of the performance achieved by running the larger model independently with the same budget.

Our results highlight the potential of using smaller models instead of larger ones, a practice that has many benefits. First, small models are far computationally cheaper to pre-train.[2] Further, at inference time, they are considerably more hardware-friendly: a 13B model can be accommodated on a single A100 GPU, a feat unachievable for a 70B model (Dettmers et al., 2022). Finally, as we have shown, when controlling for the compute budget, smaller models may actually outperform larger ones.

Our findings also emphasize the importance of developing effective ranking approaches for LLM outputs. This is especially important in cases where no unit-tests or other verification methods are available (Zou et al., 2021; Uesato et al., 2022; Sun et al., 2023). To support such research direction, we release 2,000 Code Llama 7B outputs for each example in HumanEval and MBPP—a total of more than 1M outputs.[1]

## 2 Evaluation under Compute Restrictions

To study our main research question—what is the optimal way of using a given LLM compute budget—we consider a code-generation setup with unit-tests (Chen et al., 2021; Austin et al., 2021; Hendrycks et al., 2021). Below we discuss our methodology for code generation evaluation under computational restrictions. We begin by describing pass@*k* (Kulal et al., 2019), the current main approach for evaluating code generation tasks (Section 2.1). We then transition to describe our variant of code generation metrics under computational restrictions (Section 2.2).

---

[2]E.g., Llama-2 7B was $\approx 10X$ faster to pre-train compared to the 70B variant (Touvron et al., 2023).

## 2.1 Standard Code Generation Evaluation

To evaluate LLM code-generation abilities, a common setup assumes a set of coding questions, each with a set of unit-tests. The LLM is fed with each question, and a fixed number of output generations (labelled $k$) are sampled. The evaluation protocol considers each question for which at least one output passes all unit-tests as correct. To estimate the performance of a model that generates $k$ outputs, it is common to generate a larger number of outputs $n$ ($> k$) and compute:

$$\text{pass@}k := \mathop{\mathbb{E}}_{\text{Problems}} \left[ 1 - \frac{\binom{n-c}{k}}{\binom{n}{k}} \right], \tag{1}$$

where $c \leq n$ is the number of examples that pass the unit-tests. The above mentioned metric results in an unbiased estimator as was shown by Chen et al. (2021).

## 2.2 Comparing LLMs of Different Sizes under a Fixed Budget

Our goal is to compare between LLMs of different sizes under a fixed compute budget. To do so, we allow smaller models, which consume fewer resources, to generate more outputs. This results in models of different sizes requiring roughly the same amount of compute.

We consider two types of compute budgets: the number of FLOPs and wall-time. For each type, a specific resource limit is set (e.g., 10k Tera-FLOPs or 8 seconds), and the model generates examples up to the point where the compute limit is reached. That is:

$$\text{pass}_{\text{flops}}@f := \text{pass@}k \quad \text{where: } k = \max_{\text{flops}(k') \leq f} k', \tag{2}$$

$$\text{pass}_{\text{time}}@t := \text{pass@}k \quad \text{where: } k = \max_{\text{time}(k') \leq t} k', \tag{3}$$

where flops($k$) and time($k$) are functions that return the FLOPs/wall-time usage of a given model that generates $k$ outputs. Notably, the FLOPs restriction is a more theoretical computational restriction, as it assumes perfect utilization of the hardware. On the other hand, the wall-time restriction is more realistic, but is hardware specific, and thus not directly comparable across different machines.

# 3 Experimental Setup

In this section we describe our experimental setup, focusing on the code benchmarks used (Section 3.1), our metrics (Section 3.2), and our experiments (Section 3.3).

## 3.1 Benchmarks

We experiment with three python code benchmarks: HumanEval (Chen et al., 2021), MBPP (Austin et al., 2021) and APPS (Hendrycks et al., 2021). The HumanEval benchmark consists of 164 function declarations alongside their documentation. The Code-LLM's task is to complete the function according to the provided documentation. MBPP consists of 500 test examples, each one is an instruction for a code function. Here, the Code-LLM is required to generate the full function. Lastly, the test subset of APPS is composed of 5k programming problems at various levels of difficulty: introductory (1k), interview (3k) and competition (1k). In the APPS tasks, the Code-LLM is required to generate the complete python file, which includes import declarations, class definitions, and so on.

## 3.2 Metrics

Computing the $\text{pass}_{\text{flops}}@f$ and $\text{pass}_{\text{time}}@t$ metrics requires an estimation of the flops($k$) and time($k$) functions from Equations (2) and (3). To estimate FLOPs usage, we use the calflops library (xiaoju ye, 2023), with input sequence length of 128. We measure wall-time while

Table 1: Code Llama FLOPS and wall-time usage per model size, along with normalized values with respect to the 7B model.

| Model Size | FLOPs (Teras) | FLOPs (norm.) | wall-time (seconds) | wall-time (norm.) |
|---|---|---|---|---|
| 7B | 1.69 | 1.00 | 395 | 1.00 |
| 13B | 3.29 | 1.95 | 667 | 1.69 |
| 34B | 8.58 | 5.08 | 2,994 | 7.58 |
| 70B | 17.60 | 10.41 | 5,605 | 14.19 |

assuming optimal throughput utilization of the hardware. Specifically, we use a node of 8 A100 GPUs, optimize the batch size per model and measure the time it takes each model to generate a subset of $\approx$1k examples from our datasets. We report the Code Llama results in Table 1, for readability we also report the normalized factor with respect to the 7B model.[3]

### 3.3 Experiments

We experiment with the Code Llama family (Roziere et al., 2023), a finetuned version of Llama (Touvron et al., 2023). Code Llama comes in various sizes, which we use for our experiments: 7B, 13B, 34B and 70B. For the smaller benchmarks, HumanEval and MBPP, we also consider the Llama-3 family (8B and 70B).

We follow Roziere et al. (2023), and use a zero-shot setting for HumanEval, a 3-shot prompting strategy for MBPP and 2-shot prompts for APPS, and limit the generation length to 512/256/256 tokens for HumanEval/MBPP/APPS. For the sampling process, we use nucleus sampling (Holtzman et al., 2019) with top-$p = 0.95$ and a temperature of 0.8/0.8/0.6 for HumanEval/MBPP/APPS, with all models sizes (Roziere et al., 2023). Finally, we also report, the pass@1 results using a greedy decoding method for all models.

To compare models in varying sizes, we select the maximal number of generations for each model with respect to the values in Table 1. Specifically, for the smaller benchmarks, HumanEval and MBPP, we generate $n = 2,000/1,000/400/200$ answers for the 7-8B/13B/34B/70B models, respectively. For the larger benchmarks, the three splits of APPS, we use $n = 1,000/500/200/100$. To get a robust estimation of these measures, we follow Chen et al. (2021) and Roziere et al. (2023), and report for all benchmarks a maximal value of $k = \frac{n}{2}$ for the pass@$k$ metric, while using all available unit-tests.

## 4 Small Models Outperform Large Ones under a Fixed Compute Budget

Results for **HumanEval** and **MBPP** using the Code Llama models are presented in Figures 2 and 3, respectively.[4] The corresponding results for the Llama-3 models can be found in Figures 10 and 11 (Appendix A). We first note that, as expected, the pass@$k$ metric improves both with model scale, and with the number of generations $k$ (sub-figure (a) in all figures). However, perhaps surprisingly, when considering the pass$_{\text{flops}}$@$f$ and pass$_{\text{time}}$@$t$ metrics (sub-figures (b) and (c)), we see a different trend—given a fixed compute budget, smaller models yield better results than larger ones. Specifically, the 7B/8B/13B models outperform the larger models across all compute budgets. Particularly, in the small budget regime (up to 32 normalized FLOPs units and 64 wall-time units) the performance gap reaches 5—15%.

Another way of looking at our results is by observing that smaller models match the performance of larger ones using substantially lower budgets. For instance, in HumanEval, the Code Llama 7B and 13B models achieve a score of 60% using one quarter of the time it takes the larger models to reach that score. This efficiency gap further increases with the

---

[3]Llama-3 8B/70B presents similar usage to Code Llama 7B/70B, with a difference of up to 7%.
[4]Tables 2 and 3 in Appendix B presents detailed results.

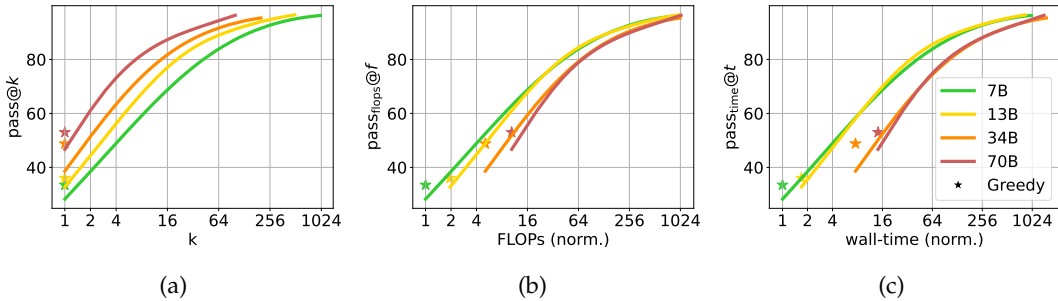

Figure 2: Code Llama performance (Y axis) as a function of compute (X axis, in exponential scale) for the HumanEval benchmark. Larger models perform better in general (Figure 2a), but under a fixed compute budget (Figures 2b and 2c), smaller models (7B and 13B) substantially outperform larger ones (34B and 70B). Greedy decoding is marked by a star.

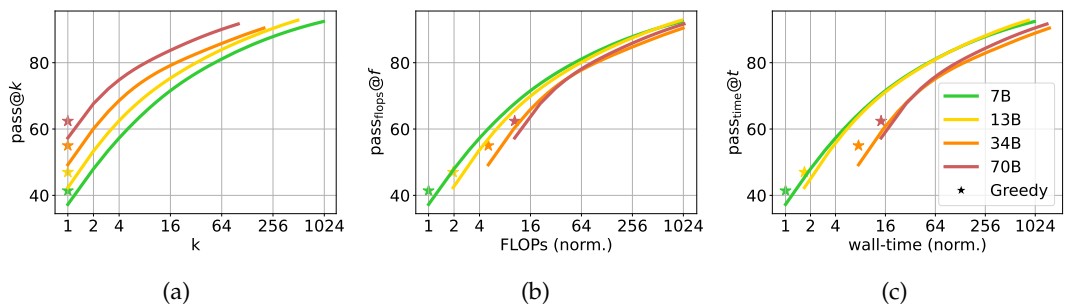

Figure 3: Code Llama performance vs. compute for the MBPP benchmark. As in HumanEval (Figure 2), larger models perform better as a function of $k$ (Figure 3a), but worse under a fixed compute budget (Figures 3b and 3c).

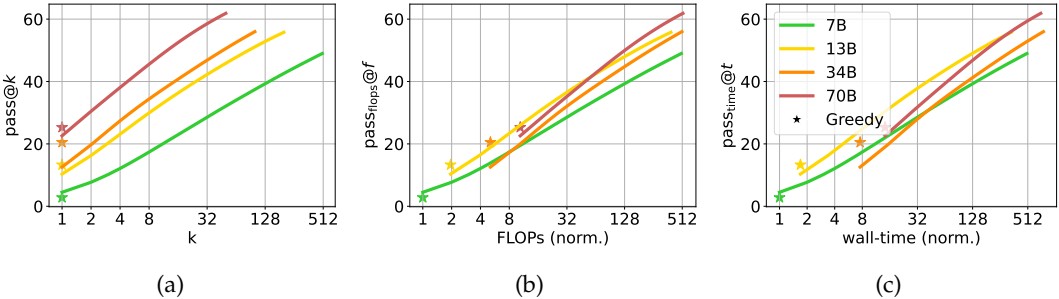

Figure 4: Code Llama performance vs. compute for the APPS benchmark, **introductory** split. The 13B model is superior to the 34B model and comparable to the 70B model under fixed budget. In contrast, the 7B model underperforms the larger models.

Llama-3 models (Figure 10c). Finally, we compare small models to greedy decoding with larger models, which generally performs better than sampling. We observe that even in this setup, using the smaller models several times is equivalent or preferable in all cases.

We next turn to discuss the Code Llama results over the three splits of the **APPS** benchmark (Figures 4 to 6).[5] We first consider the 13B model, and observe the same trends as in HumanEval and MBPP: this model achieves the best performance in almost all fixed compute budgets. Specifically for the competition split (Figures 6b and 6c), the most challenging APPS split, the 13B model outperforms all other models in all compute budgets, with a consistent margin of ≈5% from the 70B model when considering the wall-time budget. We

---

[5]Table 4 in Appendix B presents detailed results.

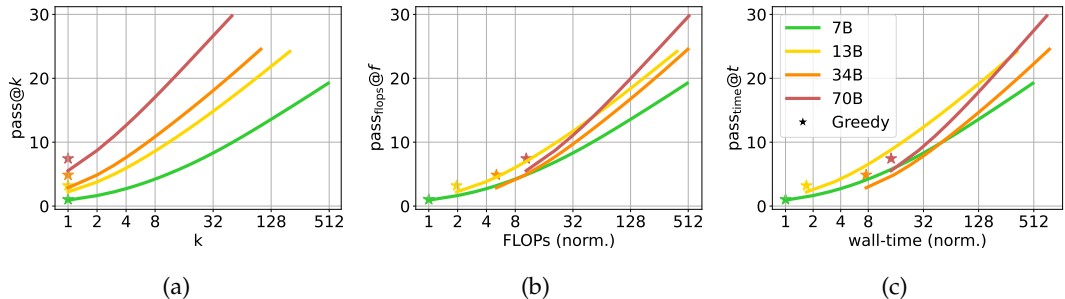

(a)                                   (b)                                   (c)

Figure 5: Code Llama performance vs. compute for the APPS benchmark, **interview** split. Similarly to the introductory split (Figure 4), the 13B model is superior to the 34B model and comparable to the 70B model under fixed wall-time, while the 7B model is inferior to the larger models.

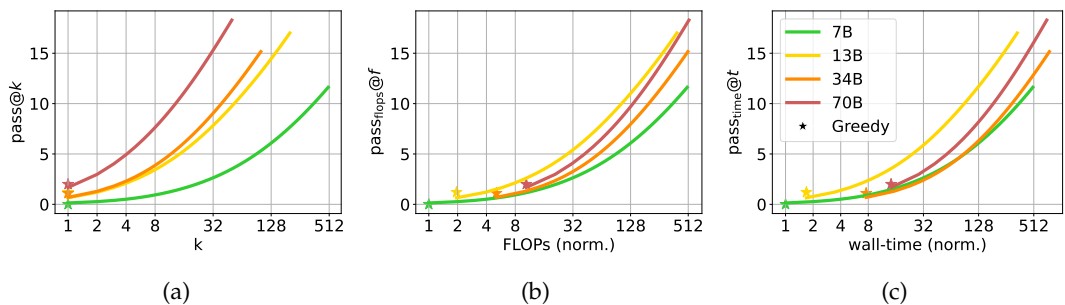

(a)                                   (b)                                   (c)

Figure 6: Code Llama performance vs. compute for the APPS benchmark, **competition** split (the most challenging one). The 13B model is superior to both 34B and 70B models under fixed wall-time, and comparable to the 70B under fixed number of FLOPs.

further observe that the 13B model achieves similar or better performance as the greedy approach of all models in all three splits. Finally, when fixing the performance, the 13B model is 2–4 times more efficient than the 70B model (both for FLOPs and wall-time).

We next observe that the 7B model is also competitive with larger models in small budget regimes (up to 8 normalized FLOPs units and 16 wall-time units). Nonetheless, it slightly underperforms the other models on larger budgets. This can be attributed to the 7B model's inability to generate a sufficient number of correct answers for the task, and may suggest that there is a minimum size requirement for a certain level of task difficulty.

Our results indicate that small models can match or even outperform large ones under a fixed compute budget, assuming the availability of unit-tests. An intriguing aspect of our research question is what happens when unit-tests are *unavailable*, and a single selection among several generations must be made. We delve into this topic in the following section.

## 5 Evaluating Code Generation without Unit-tests

We examine the scenario where unit-tests are not available (e.g., IDE code-completion setup). In this case, an efficient selection policy strategy may be used to select one answer from the model's generations. In the previous cases (Section 2), unit-tests served as this policy. Here we investigate using ranking as a selection policy. In Section 5.1 we show how to estimate the performance of a model given such a strategy, and in Section 5.2 we analyze the performance of larger models as rankers for a small model.

```
1  def rank_score_at_k(n, k, pass_sorted):
2      """
3      :param n: total number of samples
4      :param k: k in rank-score@k
5      :param pass_sorted: a binary list of pass scores. The list is
       sorted by the ranks assigned to examples by a ranker.
6      """
7      numerator_sum = 0
8      for i in range(1, n-k+2):
9          numerator_sum += math.comb(n-i, k-1) * scores_and_pass[i-1]
10     score = (numerator_sum / math.comb(n, k)) * 100
11     return score
```

Figure 7: A Python implementation of rank-score@*k* as presented in Equation (4).

## 5.1 Evaluating Rankers

We assume a model that generates *k* outputs, and a policy that ranks them. To estimate the performance of such setup, we count the number of groups containing *k* generations where the highest-ranked generation within them is a correct one. That is:

$$\text{rank-score@}k := \mathop{\mathbb{E}}_{\text{Problems}} \left[ \frac{1}{\binom{n}{k}} \cdot \left( \sum_{i=1}^{n-k+1} \binom{n-i}{k-1} \cdot \text{pass}_i \right) \right], \tag{4}$$

where $n(>k)$ is the number of answers generated for the estimation, and $[\text{pass}_1, \text{pass}_2, \dots, \text{pass}_n] \in \{0,1\}^n$ are the pass scores sorted according to the ranking policy. That is, $\text{pass}_i$ is 1 if the example ranked $i$ according to the policy is correct, and 0 otherwise. See Figure 7 for a python implementation of rank-score@*k*.

Similarly to Equations (2) and (3), we also define:

$$\text{rank-score}_{\text{flops}}@f := \text{rank-score@}k \quad \text{where: } k = \max_{\text{flops}(k') \leq f} k', \tag{5}$$

$$\text{rank-score}_{\text{time}}@t := \text{rank-score@}k \quad \text{where: } k = \max_{\text{time}(k') \leq t} k', \tag{6}$$

where flops($k$) and time($k$) are the same functions as in Section 2.2. Next, we evaluate the performance of large models as rankers using the above metrics.

## 5.2 Large Language Models as Rankers

We examine the usage of LLMs as rankers. To produce a ranking order over a set of generations, we use the averaged Negative Log Likelihood (NLL) the LLM assigns to each generation (excluding the prompt), and rank the generations according to that score. It should be noted that extracting the NLL of a model over a given generation can be done in a parallel manner (i.e., non-autoregressively), which is substantially faster than traditional token-by-token generation. The score given by a model to a generation $G = (w_1, \dots, w_l)$ given a prompt $P$ is:

$$\text{score}_{model} = \text{NLL}_{model}(G|P) = -\frac{1}{l} \sum_{i=1}^{l} \log \left( p_{model}(w_i | w_{i-1}, \dots, w_1, P) \right). \tag{7}$$

To study the performance of LLMs as rankers we use the HumanEval and MBPP benchmarks. We use 2,000 generations produced by Code Llama 7B as described in Section 3.3. As rankers we use all four Code Llama model sizes. We discard any generation that fails to complete, i.e. reached the maximal number of generated tokens without producing an end-of-sequence token. We also report the performance of running each model independently with one generation budget (both greedy and sampling).

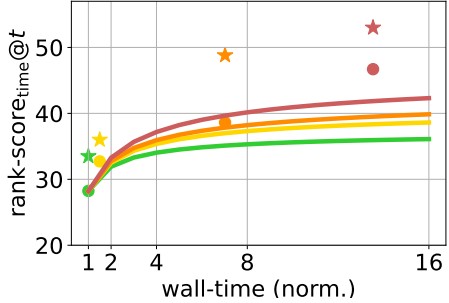 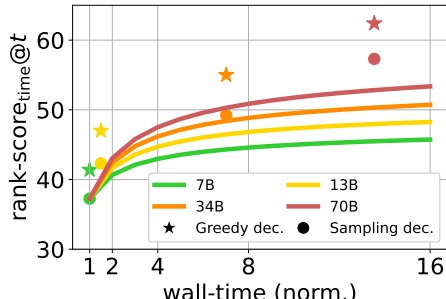

Figure 8: rank-score$_{\text{time}}$@$t$ as a function of wall-time for HumanEval (left) and MBPP (right), using different rankers (different lines). Greedy sampling is marked as a star, and top-p sampling as a circle. While ranking results improve with the size of the ranker and with compute budget, they still fall short of greedy decoding with larger models.

Our results are presented in Figure 8. As can be seen, using LLMs as rankers over generations obtained from smaller models improves performance. Interestingly, we observe that using a 7B model as a ranker for itself can enhance its generation even further than the greedy approach, albeit with the cost generating several outputs. We also find that using larger models as rankers results in better perfomance. When considering a fixed compute budget, we find that it is sometimes comparable to use LLMs as rankers instead of sampling from them, as can be seen with the 13B and 34B models. However, this is not the case for the greedy approach which consistently outperforms ranking multiple generations from a smaller model given a fixed compute budget.

To further check the use of external verifiers, we integrate the LEVER verifier model (Ni et al., 2023) with the Code Llama models. The LEVER approach aims to enhance code generation by learning to verify generated programs. The full LEVER pipeline involves using the NLL produced by the code generation model, error pruning based on execution, and a verifier trained on code generations with execution results. However, since we assume that no tests are available in our setting, execution pruning and execution results cannot be used. LEVER released a trained verifier over the MBPP benchmark, which we use along with the NLL scores of each model. As shown in Figure 9, the LEVER verifier does not improve the results in the test-less setting, which is expected given that one of the main components of the approach relies on execution over unit-tests.

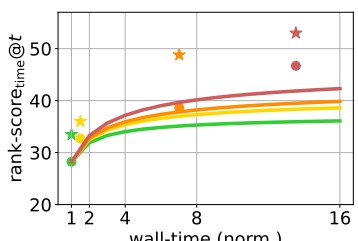

Figure 9: rank-score$_{\text{time}}$@$t$ as a function of wall-time for MBPP, using the LEVER verfier with different NLL rankers. Results are similar to Figure 8.

In summary, there remains a gap to bridge between using LLMs as rankers for smaller models and using them as generators. To further promote this line of research, we release the 2,000 generations per example produced by the 7B model for both HumanEval and MBPP (a total of 1,328,000 generations).

## 6   Related Work

### 6.1   Model Scaling

Model scaling was found to be one of the key elements in the success of LLMs (Dehghani et al., 2023; Gu et al., 2023; Hassid et al., 2023; Rae et al., 2021; Chowdhery et al., 2023; Touvron et al., 2023), with Wei et al. (2022) demonstrating how specific abilities emerge mainly after reaching a specific scale. The way language models behave when they are scaled up and their ability to adjust have been a significant factor in the creation of LLMs (Hernandez et al., 2021). Kaplan et al. (2020) investigated the optimal model size to train for a given

compute budget, while Hoffmann et al. (2022) demonstrated how scaling both model and dataset sizes improves performance across various tasks. Clark et al. (2022) analyzed the scaling properties of mixture-of-experts models, showing that scaling with the number of experts diminishes as model size increases. Recently, Gadre et al. (2024) provided a scaling law analysis considering downstream tasks rather than next-token prediction loss. They related the perplexity of a language model to its downstream task performance via a power law and used it to predict the top-1 error averaged over the evaluated downstream tasks. Our work differs from all of the above, as we do not claim to provide new scaling laws but rather suggest that when fixing the budget, smaller models can provide comparable or superior results to larger ones.

Recent studies by Shi et al. (2024) and Mei et al. (2024) have demonstrated that under constrained compute budgets, smaller vision models can surpass their larger counterparts. Specifically, Shi et al. (2024) found advantages in using multiple image scales, whereas Mei et al. (2024) observed that smaller diffusion models perform better than larger ones when the compute budget is fixed. Our approach, which generates multiple text outputs from a small model, aligns with these findings.

## 6.2 Verifiers and Rankers

LLM verifiers and rankers is a growing trend, which leverages LLMs to verify and rank generations obtained from weaker and smaller models (Cobbe et al., 2021b; Uesato et al., 2022; Saha et al., 2024; Havrilla et al., 2024). Both Cobbe et al. (2021b) and Uesato et al. (2022) leveraged an external classifier to rank LLM outputs. Specifically, in both setups the authors proposed to generate many candidate solutions and select the one ranked highest by the verifier. The authors demonstrated the applicability of using such verifiers in solving math word problems (Cobbe et al., 2021a). Qin et al. (2023) demonstrated that LLMs can serve as efficient text rankers when considering pairwise ranking.

Another line of work leveraged LLMs to evaluate the quality of smaller models (Saha et al., 2024; Dubois et al., 2023; Zheng et al., 2023; Oren et al., 2024). Although providing a promising alternative, such evaluation suffers from biases in the larger model (Zheng et al., 2023) and reliance on hand-designed evaluation plans that impact the method's ability to generalize (Liu et al., 2023). Large models also serve as verifiers of small ones in a speculative decoding setup, with the goal of speeding-up LLM generation (Leviathan et al., 2023; Kim et al., 2023; Chen et al., 2023). It is also common to distill knowledge from a large model into a smaller one in order to improve efficiency (Hinton et al., 2015; Sanh et al., 2019; Xu et al., 2024), see Treviso et al. (2023) for a survey on efficient methods in NLP.

In this work, we explore the potential of LLMs as selectors of the best output of a smaller model in a fixed budget setup. Similarly to ours, Li et al. (2024) found that smaller sized LMs (7B parameters) already exhibit strong mathematical abilities when selecting the best response from $k$ different generations. When considering code generation models, AlphaCode Team (2023) presented impressive results on challenging coding contests tasks while generating 1M samples, and later on filtering and ranking them using Gemini-Pro LLM (Team et al., 2023). Dou et al. (2024) proposed a method to improve code-generation models by learning a policy model using reinforcement learning methods. Lastly, Shi et al. (2022) and Ni et al. (2023) used execution feedback in order to filter code-generations, while Shi et al. (2022) used non-learned approaches, Ni et al. (2023) trained an external verifier on top of the generation and the execution feedback.

# 7 Discussion & Limitations

Our results show that using smaller models with the same amount of compute can improve LLM code-generation performance. An interesting question we do not fully address is whether, given enough compute, the larger models will overtake the smaller ones, or perhaps they will all saturate at a similar performance level at some point. Our HumanEval and MBPP results seem to slightly support the latter hypothesis (as all models begin to saturate, see Figures 2 and 3). However, unfortunately, due to compute constraints, our

setting is restricted to exploring only a limited number of generations per model.[6] We note that despite this limitation, in practice, due to these costs our conclusions apply to most practical use-cases. We defer more expensive experiments to future work.

# 8  Conclusion

In this work, we compared large language models with smaller-sized models under fixed budget constraints (i.e., FLOPs and wall-time). We evaluated the models using execution-based code-generation tasks, which provide access to unit-tests. Our findings reveal that generating multiple outputs from a 13B model may lead to gains of up to 15% over a single generation from a 70B model across five tasks. This highlights the potential of using smaller models instead of larger ones. In scenarios where unit tests or other solution verifiers are unavailable, we explored a simple ranking-based approach for candidate selection. We found the proposed ranking approach falls short in performance compared to a single output from the larger model. Our findings emphasize the importance of studying approaches for ranking LLM outputs, which hold great potential to not only improve model performance but also improve budget allocation. To further enhance this research direction we release over 1M samples from the Code Llama 7B models spanning both HumanEval and MBPP benchmarks.

# 9  Acknowledgments

We thank Miri Varshavsky Hassid for the great feedback and moral support.

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

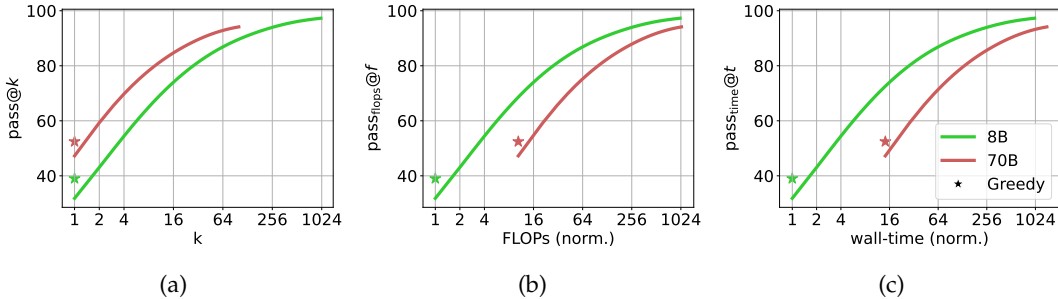

Figure 10: Llama-3 performance vs. compute for the HumanEval benchmark. The 70B model performs better in general (Figure 10a), but under a fixed compute budget (Figures 10b and 10c), the 8B model substantially outperforms the larger one.

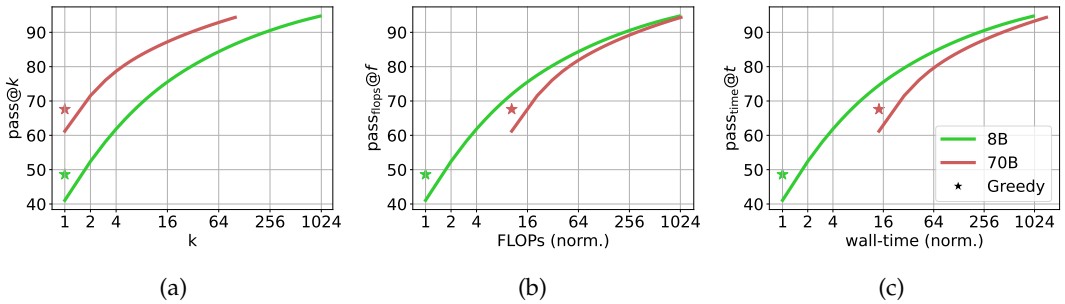

Figure 11: Llama-3 performance vs. compute for the MBPP benchmark. As in HumanEval (Figure 10), larger models perform better as a function of $k$ (Figure 11a), but worse under a fixed compute budget (Figures 11b and 11c).

## A  Llama-3 Results

We present Llama-3 results for the HumanEval and MBPP benchmarks in Figures 10 and 11, respectively.

## B  Detailed pass@$k$ Results

In Tables 2 to 4 presents precise pass@$k$ results for the datasets examined (HumanEval, MBPP and APPS, respectively). Due to the infeasibility of reporting results for all $k$, we provide results for selected $k$ values. Nevertheless, it is important to note that all relevant $k$ values were calculated and used in the computation of the figures.

Table 2: Precise models' pass@$k$ results for several $k$ values over the HumanEval benchmark.

| $k$ for pass@$k$ 
 Model | 1 | 2 | 4 | 16 | 64 | 128 | 256 | 500 | 1000 |
|---|---|---|---|---|---|---|---|---|---|
| Code Llama 7B | 28.2 | 38.5 | 48.9 | 68.7 | 83.9 | 89.0 | 92.7 | 95.0 | 96.3 |
| Code Llama 13B | 32.7 | 44.4 | 56.2 | 77.0 | 89.0 | 92.3 | 94.8 | 96.5 | –.– |
| Code Llama 34B | 38.6 | 51.4 | 63.4 | 81.8 | 91.7 | 94.3 | –.– | –.– | –.– |
| Code Llama 70B | 46.7 | 61.1 | 73.2 | 87.3 | 94.4 | –.– | –.– | –.– | –.– |
| Llama-3 8B | 31.8 | 43.1 | 54.4 | 74.0 | 86.9 | 90.9 | 94.0 | 96.0 | 97.3 |
| Llama-3 70B | 47.3 | 59.3 | 69.8 | 84.7 | 92.8 | –.– | –.– | –.– | –.– |

Table 3: Precise models' pass@*k* results for several *k* values over the MBPP benchmark.

| *k* for pass@*k*  Model | 1 | 2 | 4 | 16 | 64 | 128 | 256 | 500 | 1000 |
|---|---|---|---|---|---|---|---|---|---|
| Code Llama 7B | 37.3 | 48.0 | 57.3 | 71.6 | 81.1 | 84.8 | 87.9 | 90.3 | 92.4 |
| Code Llama 13B | 42.3 | 53.4 | 62.5 | 75.3 | 84.0 | 87.4 | 90.4 | 92.8 | –.– |
| Code Llama 34B | 49.2 | 60.1 | 68.6 | 79.2 | 85.8 | 88.7 | –.– | –.– | –.– |
| Code Llama 70B | 57.3 | 67.7 | 74.8 | 83.7 | 90.0 | –.– | –.– | –.– | –.– |
| Llama-3 8B | 41.1 | 52.4 | 61.8 | 75.5 | 84.4 | 87.7 | 90.5 | 92.8 | 94.7 |
| Llama-3 70B | 61.2 | 71.5 | 78.7 | 87.2 | 92.9 | –.– | –.– | –.– | –.– |

Table 4: Precise models' pass@*k* results for several *k* values over the different splits of the APPS benchmark.

| *k* for pass@*k*  Model | 1 | 2 | 4 | 16 | 32 | 64 | 128 | 256 | 500 |
|---|---|---|---|---|---|---|---|---|---|
| APPS-introductory | | | | | | | | | |
| Code Llama 7B | 4.5 | 7.7 | 12.1 | 23.0 | 28.6 | 34.0 | 39.3 | 44.4 | 49.0 |
| Code Llama 13B | 10.3 | 16.3 | 23.1 | 36.3 | 42.3 | 47.8 | 52.8 | –.– | –.– |
| Code Llama 34B | 12.6 | 19.7 | 27.4 | 40.8 | 46.8 | 52.5 | –.– | –.– | –.– |
| Code Llama 70B | 22.6 | 30.5 | 38.1 | 52.5 | 58.5 | –.– | –.– | –.– | –.– |
| APPS-interview | | | | | | | | | |
| Code Llama 7B | 0.9 | 1.6 | 2.7 | 6.1 | 8.3 | 10.8 | 13.6 | 16.4 | 19.2 |
| Code Llama 13B | 2.2 | 3.8 | 5.9 | 11.6 | 14.8 | 18.3 | 21.9 | –.– | –.– |
| Code Llama 34B | 2.9 | 4.9 | 7.6 | 14.4 | 18.1 | 22.0 | –.– | –.– | –.– |
| Code Llama 70B | 5.5 | 8.7 | 12.6 | 21.8 | 26.6 | –.– | –.– | –.– | –.– |
| APPS-competition | | | | | | | | | |
| Code Llama 7B | 0.1 | 0.3 | 0.5 | 1.6 | 2.6 | 4.1 | 6.1 | 8.6 | 11.6 |
| Code Llama 13B | 0.6 | 1.2 | 2.1 | 5.3 | 7.8 | 10.9 | 14.4 | –.– | –.– |
| Code Llama 34B | 0.7 | 1.3 | 2.3 | 6.1 | 9.0 | 12.6 | –.– | –.– | –.– |
| Code Llama 70B | 1.7 | 3.0 | 4.9 | 11.1 | 15.3 | –.– | –.– | –.– | –.– |

