# OpenReview forum: "The Larger the Better? Improved LLM Code-Generation via Budget Reallocation"
_colmweb.org/COLM/2024/Conference — COLM_

### Official Review · Reviewer_NH43 · 2024-05-11

**Rating:** 5
**Confidence:** 2
**Ethics Flag:** 1

**Summary:**

The paper compares large and smaller LMs for the code generation under the same computation budget (FLOPs or wall-time). There are two setups in the paper depending on the availability of unit tests. When the unit tests are available, generating more outputs using smaller LLMs increases the likelihood of getting a correct answer. When the unit tests are unavailable, greedy decoding with large LMs seems to be a good choice.

The writing of the paper is clear and self-contained. As someone who is not familiar with code generation, I find the paper is easy to follow. The idea of turning the evaluation from pass@k to pass@flop and pass@time is simple but well-motivated given the current methodology of evaluation with pass@k. The paper executes the simple idea well.

**Reasons To Accept:**

- The paper could be interesting for code generation community
- The work is incremental but solid

**Reasons To Reject:**

- While we see the relative improvement of smaller LMs in pass@flop and pass@time, the absolute value of pass@flop (and also pass@k) seems to indicate that current Code Llama is far from usable. Thus I felt that the recommendation of using smaller LLMs to generate multiple outputs is not as critical as improving code generation LLMs itself.
- The paper mentioned the common practice of scaling LLMs as a sole drive for improving performance. This is not true. In the past years the community has advocated for better pretraining/fine-tuning data. Thus I feel that the research in the paper is not far ahead looking.
- In practice, there are multiple ways to improve generation speed/quality. For example, applying speculative decoding to LLM or distilling large ones to smaller ones. Those common practices are not studied in the paper thus it makes the paper have less practical value.

---

> ### Author Rebuttal · Authors · 2024-05-30
>
> We thank the reviewer for their insightful comments. We appreciate the reviewer for thinking that our evaluation metrics are “well-motivated”, noting that we execute our ideas well and acknowledging that the paper is “self-contained” and “easy to follow”.
>
> Regarding the absolute performance of the Code Llama models family, we agree that better results would be advantageous. However, we would like to point out that despite this, current code generation models have already shown practical utility, as evidenced by the wide adoption of tools like co-pilot, the cursor IDE, etc. by software developers. As a result, reducing their computational costs is of great practical value.
>
> Regarding scaling as a means to improve performance, we certainly acknowledge the reviewer's point and wish to clarify that we do not claim that scaling is the sole method for enhancing LLMs. Our statement was intended to highlight that scaling up is a common approach, though not the only one, currently used to improve models. We will make this distinction clearer.
>
> Regarding other ways to improve generation speed/quality, we recognize that there are various strategies to improve the performance of LLMs. However, we would like to emphasize that our paper presents a unique perspective by focusing on the efficient use of LLMs within specific budget constraints. This approach is orthogonal to other generation speed/quality methods, and we believe it contributes uniquely to the field.

---

> > ### Author Response · Authors · 2024-06-05
> >
> > We would like to thank you once again for taking the time to review our manuscript! Your comments and feedback are highly appreciated.
> >
> > The discussion period is about to end, and we wanted to ask whether there are any more details we can clarify about our work.

---

### Official Review · Reviewer_52pF · 2024-05-11

**Rating:** 5
**Confidence:** 3
**Ethics Flag:** 1

**Summary:**

The paper investigates whether smaller LLMs can outperform or match the performance of larger models when constrained by the same computational budget. The authors present a approach that utilizes smaller models multiple times within a given budget and selects the best output using a ranking method. This study is situated within the broader discourse on the efficiency of model scaling and the practical constraints of deploying large LLMs.

**Reasons To Accept:**

The paper reveals that smaller LLMs can outperform or match the performance of larger models when constrained by the same computational budget. The findings have practical implications, especially for applications with limited computational resources.

**Reasons To Reject:**

1. In practical applications, developers aim to reduce computational costs while enhancing performance. However, the paper only presents results under a constant computational budget. It would be beneficial for the study to explore ways to improve performance across various computational budgets.
2. The reliability of using large language models as rankers, as proposed in section 5.2, is questionable. The authors should conduct further analysis on this aspect.
3. In the related work section, the authors should also discuss the direction for enhancing generation speed and quality with smaller models (such as distillation).

---

> ### Author Rebuttal · Authors · 2024-05-30
>
> We thank the reviewer for their meaningful review. We are encouraged to hear that the reviewer found that our findings “have practical implications, especially for applications with limited computational resources”.
>
> Addressing specific reviewer comments:
>
> 1. Regarding practical applications, we agree that it would be beneficial to study ways to improve performance across various compute budgets. However, we also like to emphasize that having a fixed budget (e.g., as a maximum cost per interaction) and a fixed model (e.g., using an open-source LLM checkpoint or API) is a very practical scenario. We also note that all our results are across various (fixed) compute budgets, and that our improvements are across the board. As a result, we believe our study provides valuable insights for the use of available resources.
>
> 2. We agree with the reviewer that ranking LLMs outputs is an interesting research area to explore. Although this is not the main contribution of our paper, we explore the use of LLMs as verifiers using their NLL score for ranking (and find that it is insufficient).
> As suggested by P86J, we tried using the recent LEVER [1] ranking approach over the generations produced by the Code Llama 7B model for the MBPP test set (using the trained model from [1]). The LEVER approach uses the LM probability over the generation combined with a probability given by a trained ranker.*
> Results are shown in Figure 3 in tinyurl.com/ykbappvj, and show a very similar trend to our experiments with NLL scores (Figure 8 in the paper). We will add this result to the paper.
> Moreover, our work sets the environment for further research in the field, both by providing code for rank-score@k metric and by releasing (upon publication) more than 1M generation examples of CodeLlama-7B for the HumanEval and MBPP datasets.
>
> * We didn’t use execution feedback, as in our **ranking** setting we assume tests are not available.
>
> 3. Thank you for your comment about discussing other methods for reducing costs. We will ensure that this topic is covered in the next version of our paper.
>
> [1] arxiv.org/abs/2302.08468

---

> > ### Author Response · Authors · 2024-06-05
> >
> > We would like to thank you once again for taking the time to review our manuscript! Your comments and feedback are highly appreciated.
> >
> > The discussion period is about to end, and we wanted to ask whether there are any more details we can clarify about our work.

---

> ### Comment · Reviewer_52pF · 2024-06-05
>
> Thanks for your response. I maintain the same scoring, and look forward to your revised version.

---

### Official Review · Reviewer_P86J · 2024-05-13

**Rating:** 7
**Confidence:** 5
**Ethics Flag:** 1

**Summary:**

This paper studies an interesting questions of given the same budget of compute or wall-time, will the smaller models outperform larger models given their lower inference cost. Experiments are conducted with the CodeLLaMA models ranging from 7B to 70B, on three function-level code generation benchmarks, HumanEval, MBPP and APPS. Results show that when test cases are available, smaller CodeLLaMA models consistently outperforms larger models. However, when test cases are not available, larger models still perform better even when likelihood-based ranking methods are used.

**Questions To Authors:**

Q1. Do you plan to add the results for at least another set of models, either code-specific (e.g., StarCoder) or general-purpose (e.g., LLaMA)?
Q2. Have you tried other reranking-based methods for code, such as MBR-Exec (Shi et al., 2022) and LEVER (Ni et al., 2023)? Also note that there are a couple of other heuristic-based reranking methods described in Ni et al., 2023. If not, can you comment on what the potential performance implications will be?
Q3. Can you comment on how methods like quantization and flash-attention would affect the FLOPs and wall-time calculated in Table 1?
Q4. For larger models, have different model parallelism methods been tried to ensure that maximal efficiency is achieved given the same compute infrastructure?
Q5. How are the tests being used for the results reported in Section 4? For example, for MBPP there are 3 test cases, do you use all three to filter the incorrect samples, or just 1?

**Reasons To Accept:**

S1. The idea of this paper is very interesting, and the results potentially have great impact on how the practitioners would choose models for their specific use cases;
S2. The experiments are nicely designed, the metrics as pass_flop@k and pass_time@k make a lot of sense to me;
S3. The flow of this paper is quite smooth, which makes it very easy to follow

**Reasons To Reject:**

W1. The major weakness of this paper is that only one set of models (i.e., CodeLLaMA) is studied in this work, which makes it unclear whether the same conclusions would generalize to other code-specific models (e.g., StarCoder) or general purpose-models (e.g., LLaMA);
W2. I found it quite strange that all of the results are reported in the form of figures, and there are no concrete numbers in such figures. For reproducibility and comparison purposes, I would suggest at least putting the concrete numbers in the appendix;
W3. There are some key experiment details that are missing, see Q5 below.

---

> ### Author Rebuttal · Authors · 2024-05-30
>
> We thank the reviewer for the insightful review. We appreciate them finding our results of “great impact,” for acknowledging that our experiments “are nicely designed,” and for considering our paper as “quite smooth” and “very easy to follow”.
>
> W1+Q1: Reg. using one model family, we point that the experiments conducted required substantial computational resources, ~100 days of 8x A100 GPUs (e.g., footnote2). Nevertheless, per the reviewer’s request, we conducted additional experiments with the LLaMA-3 [2] models (8&70B) over HumanEval and MBPP, see tinyurl.com/ykbappvj, Figures 1&2. The trends mirror those seen with the Code Llama models, i.e: under a fixed compute budget, running the 8B model multiple times substantially outperforms running the 70B model fewer times. We will include these results in the paper.
>
> W2: We will add to the paper all pass@$k$ numbers.
>
> W3+Q5: All available unit-tests are used to verify the correctness of each generation, i.e., only solutions that pass all unit-tests are considered correct. We will clarify this.
>
> Q2: Reg. trying other ranking methods, although this is not our main contribution, we explore the use of LLMs as verifiers using their NLL score for ranking.
> As for the reviewer’s suggestion, we tried using the LEVER [1] ranking approach over the Code Llama 7Bs’ generations for MBPP (using the trained ranker from [1]). This approach uses a combination of the LM and the trained ranker probabilities.*
> Results are shown in Figure 3 in tinyurl.com/ykbappvj, and show a similar trend to our experiments with NLL scores (Figure 8 in the paper). We will add this result to the paper.
> Moreover, we do set the environment for further research, both by providing code for rank-score@k metric and by releasing (upon publication) more than 1M generations of Code Llama 7B for HumanEval and MBPP.
>
> * We didn’t use execution feedback, as in our ranking setting we assume tests are not available.
>
> Q3+Q4: We use a custom implementation tuned for fast inference, based on the xFormers library and its "memory_efficient_attention" primitive using FlashAttention (github.com/facebookresearch/xformers). This primitive dispatches automatically to different backends to maximally optimize inference time, based on input params, GPUs, arch., etc. We believe quantization/other efficiency methods will affect our numbers similarly for all model sizes, hence can be considered as an orthogonal direction.
>
> [1] arxiv.org/abs/2302.08468; [2] llama.meta.com/llama3/

---

> > ### Comment · Reviewer_P86J · 2024-06-03
> > **Thanks for the response**
> >
> > I would like to thank the authors for the response.
> >
> > I think all my concerns are adequately addressed, adding the results for LLaMA-3 will definitely make the paper stronger to show that the conclusions generalize to non-code-specific LLMs.
> >
> > Regarding W2 and Q3+Q4, please include these discussions in the next version of the paper as I think those would make the paper more holistic.
> >
> > I'm increasing my score based on the response, good luck!

---

> > > ### Author Response · Authors · 2024-06-04
> > >
> > > We thank the reviewer for the positive feedback and constructive comments. We will incorporate the suggested discussions in the paper.

---

### Official Review · Reviewer_ZRmm · 2024-05-14

**Rating:** 6
**Confidence:** 4
**Ethics Flag:** 1

**Summary:**

The paper explores the hypothesis of whether larger models are better than smaller models even in limited computational budget scenarios. In other words, the question the paper explores is: can we generate more outputs from a smaller model so that it matches the performance of bigger models under a certain computational budget? The analysis focuses on the task of code generation. The models are large language models (LLMs) of different sizes and the smallest size is 7B parameters while the largest is 70B parameters. The results of the paper show that repeated use of smaller models can yield consistent improvements with gains of up to 15% across different tasks. In scenarios in which a candidate generation needs to be ranked and then selected as the best one, the smaller model fall short of the performance of a single greedy decoding output from a larger model. It is unclear from the paper the reason for this result.

**Reasons To Accept:**

The paper is easy to follow, well written and well organized. The experimental settings seem robust and well designed. The paper explores an interesting and important problem that is assessing whether we actually need to use the largest model available always, under every computational budget condition. This is useful for the community that works on building such models and applications with them so that they make informed decisions when designing model architectures and approaches that rely on them.

**Reasons To Reject:**

Mainly related to the generalization of the results across different generation tasks or all types of tasks. The paper focuses solely on one task, code generation, for which there is a big interest in the software engineering community. However, it is difficult to generalize the conclusions obtained for code generation for every other generation task. The main reason is because the evaluation metrics change according to the task.

It is not completely clear and focused on the paper why the ranked-selection-based task (for code generation cases without unit tests) do not work well for smaller models. Previous work [1], [2] explored re-ranking approaches to other NLP problems such as machine translation in which the the ranker is an LLM trained with human-labeled supervision about the translation quality of translated outputs. In this scenario, re-ranking with this specialized ranker usually works well but no analysis of different LLMs sizes has been done. Perhaps this is an interesting research direction that complements the work here.

Overall the paper could use from experiments on other tasks besides code generation to be able to claim that smaller models are better than larger ones.

[1] https://arxiv.org/abs/2205.00978
[2] https://aclanthology.org/2023.emnlp-main.733/

---

> ### Author Rebuttal · Authors · 2024-05-30
>
> We thank the reviewer for their valuable feedback. We appreciate them approving that our paper “explores an interesting and important problem”, acknowledging the improvements shown in our results and our experimental setup being “robust and well designed”, stating that the paper is “well written and well organized”, and overall “useful for the community”.
>
> Regarding the comment about evaluation on other tasks: we note that the focus of this work is on tasks that have a fast and reliable way to verify the output. To the best of our knowledge, the major task with practical relevance that meets this criterion is execution based code generation, as, for example, test-driven development is a common approach in software development. Notice, under the text-to-image domain, recent studies found similar trends regarding smaller models outperform larger ones under fixed budget, see [2]. Nevertheless, we agree that demonstrating similar properties for other verifiable NLP downstream tasks is interesting, and would appreciate any recommendations for such tasks.
>
> We agree that ranking LLM outputs is an interesting research area to explore. Although this is not our main contribution, we explore the use of LLMs as verifiers using their NLL score for ranking (and find that it is insufficient).
> As the reviewer mentioned, there are studies that use trained rankers. We tried using the LEVER [1] ranking approach over the generations produced by the Code Llama 7B model for the MBPP test set (using the trained model from [1]). The LEVER approach uses the LM probability over the generation combined with a probability given by a trained ranker.*
> Results are shown in Figure 3 in tinyurl.com/ykbappvj, and show a very similar trend to our experiments with NLL scores (Figure 8 in the paper). We will add this result to the paper.
> Moreover, our work sets the environment for further research in the field, both by providing code for rank-score@k metric and by releasing (upon publication) more than 1M generation examples of CodeLlama-7B for the HumanEval and MBPP datasets.
>
> * We didn’t use execution feedback, as in our **ranking** setting we assume tests are not available.
>
> [1] arxiv.org/abs/2302.08468
> [2] arxiv.org/abs/2404.01367

---

> > ### Author Response · Authors · 2024-06-05
> >
> > We would like to thank you once again for taking the time to review our manuscript! Your comments and feedback are highly appreciated.
> >
> > The discussion period is about to end, and we wanted to ask whether there are any more details we can clarify about our work.

---

> > > ### Comment · Reviewer_ZRmm · 2024-06-07
> > >
> > > Thank you for the response, I'm keeping the scores as is.

---

### Decision · Program_Chairs · 2024-07-10

**Decision:**

Accept

**Comment:**

The paper explores whether smaller language models can perform as well as, or better than, larger models under constrained computational budgets, specifically within the context of code generation tasks. The study compares the performance of CodeLLaMA models ranging from 7B to 70B parameters, utilizing benchmarks such as HumanEval, MBPP, and APPS. The results indicate that while smaller models can achieve consistent improvements through unit tests, they fall short in ranking-based scenarios without test cases.

This paper investigates an interesting question in the field of NLP: the efficiency and performance trade-offs between smaller and larger models. The experiments are well-designed, and the findings are useful and have insights. Overall, this paper presents a well-executed study on an important topic with practical implications and I recommend acceptance. Addressing the noted weaknesses and expanding the scope of experiments would significantly enhance its contributions to the field.